# Co-Creation of Massive Open Online Courses to Improve Digital Health Literacy in Pregnant and Lactating Women

**DOI:** 10.3390/ijerph19020913

**Published:** 2022-01-14

**Authors:** Yolanda Álvarez-Pérez, Lilisbeth Perestelo-Pérez, Amado Rivero-Santanta, Alezandra Torres-Castaño, Ana Toledo-Chávarri, Andrea Duarte-Díaz, Vinita Mahtani-Chugani, María Dolores Marrero-Díaz, Alessia Montanari, Sabina Tangerini, Carina González-González, Michelle Perello, Pedro Serrano-Aguilar

**Affiliations:** 1Canary Islands Health Research Institute Foundation (FIISC), 38109 Tenerife, Spain; amado.riverosantana@sescs.es (A.R.-S.); atorrcas@sescs.es (A.T.-C.); anatoledochavarri@sescs.es (A.T.-C.); andrea.duartediaz@sescs.es (A.D.-D.); 2Evaluation Unit (SESCS), Canary Islands Health Service (SCS), 38109 Tenerife, Spain; lilisbeth.peresteloperez@sescs.es (L.P.-P.); pseragu@gobiernodecanarias.org (P.S.-A.); 3Research Network on Health Services in Chronic Diseases (REDISSEC), 38109 Tenerife, Spain; 4Center for Biomedical Research of the Canary Islands (CIBICAN), 38320 Tenerife, Spain; 5Gerencia de Atención Primaria de Tenerife, 38004 Tenerife, Spain; vmahchu@gobiernodecanarias.org (V.M.-C.); mmardial@gobiernodecanarias.org (M.D.M.-D.); 6Associazione Comitato Collaborazione Medica (CCM), 10152 Torino, Italy; Alessia.Montanari@theglobalfund.org (A.M.); sabina.tangerini@amref.it (S.T.); 7ITED Research Group, Department of Computer Science and Engineering, University of La Laguna (ULL), 38200 Tenerife, Spain; cjgonza@ull.edu.es; 8Consulta Europa Projects & Innovation, 35006 Las Palmas, Spain; michelle.perello@consulta-europa.com

**Keywords:** pregnancy, lactation, digital health literacy, health education, MOOC

## Abstract

Background: Digital health literacy (DHL) increases the self-efficacy and empowerment of pregnant and lactating women (PLW) in using the Internet for health issues. The European project IC-Health aimed to improve DHL among PLW, through the co-creation of Massive Open Online Courses (MOOCs). Methods: The co-creation of the MOOCs included focus groups and the creation of communities of practice (CoPs) with PLW and healthcare professionals aimed to co-design the MOOCs. The quantitative measures of MOOCs’ acceptability, experience in the co-creation process and increase in DHL (dimensions of finding, understanding and appraisal) were assessed. Results: 17 PLW participated in focus groups, 113 participants were included in CoPs and 68 participants evaluated the acceptability of MOOCs. A total of 6 MOOCs aimed at improving PLW’s DHL were co-designed. There was a significant improvement in self-perceived DHL after using MOOCs (*p*-value < 0.001). The acceptability of MOOCs and co-creation experience were positively valued. Conclusions: The preliminary results of the quantitative assessment showed a higher self-perceived DHL after the IC-Health MOOCs. These results suggest that IC-Health MOOCs and the co-creation methodology appear to be a viable process to carry out an intervention aimed to improve DHL levels in European PLW.

## 1. Introduction

Pregnant and lactating women (PLW) usually consult web-based sources to find information about health prevention and care [1,2,3,4]. However, many non-scientific health information websites with low-quality content [5,6] could be interpreted as reliable by the general population [7]. Furthermore, PLW do not usually discuss their online findings with their healthcare professionals [7,8,9]. Health education can have an impact on the physical and mental self-care of PLW, preventing illnesses and complications for them and their children [10,11,12,13]. Therefore, PLW are considered a relevant target population to promote health literacy (HL).

HL or digital health literacy (DHL), when digital devices are used, refer to the social and cognitive competencies to obtain, process and understand health information in order to make appropriate health decisions, irrespective of the educational level or general reading ability [14]. The skills related to DHL are to find, understand, and appraise health information from electronic sources and apply the knowledge gained to address or solve a health problem. Despite some interventions that were developed to promote the use of electronic resources to improve DHL levels in PLW with low HL [15,16,17,18], the availability of online tools is low [19] and heterogeneous [20,21].

Massive Open Online Courses (MOOCs) represent an innovative tool to provide effective, quality, equitable, and person-centered health education. MOOCs are free web-based open courses available to anyone everywhere and have the potential to revolutionize education by increasing the accessibility and reach of education to unlimited numbers of participants [22,23,24]. MOOCs usually have a learning-oriented structure, with tests or evaluations to accredit the knowledge acquired. There are several examples of MOOCs aimed to improve people’s knowledge and self-management of health issues [19,25].

Given the complexity of health information, several studies suggest using a person-centered approach to design and develop health websites and tools, involving the perspective and preferences of the target users from the initial phases of the development process [26]. This co-creation process may be an effective strategy to integrate PLW and healthcare providers’ perspectives in the design of potential solutions for increased patient self-efficacy and empowerment [27,28]. Therefore, the co-development with the target audience of a MOOC focusing on essential DHL skills [29] can be an appropriate educational tool to enhance the DHL of PLW and increase their empowerment.

The increased costs of the European Union (EU) health and social care systems led to the European Commission to identify some gaps related to DHL levels of PLW, and therefore it established a research and innovation plan that actively involved them in the improvement of their health promotion [30]. The EU has emphasized the importance of improving citizens’ DHL to take advantage of the opportunities offered by eHealth tools, and to obtain better health outcomes and safer care [31]. The European Commission’s eHealth Action Plan 2012–2020 provided a roadmap to empower patients and healthcare workers [32]. In this sense, the development of MOOCs oriented to improve the cognitive abilities underlying DHL represents a promising strategy in terms of effectiveness and efficiency [19].

In this regard, the European project IC-Health: Improving digital health literacy in Europe, aimed to improve the DHL levels of European PLW and other population cohorts, through the co-creation of MOOCs focusing on the essential DHL skills [29].

In this article, we present: (1) the results of the focus groups carried out to explore the experience of PLW in the use of the Internet for health-related issues, as well as their needs and expectations, in order to qualitatively explore the dimensions of DHL and to complement the information of the survey; (2) the co-creation methodology applied and the developed MOOCs; and (3) a pilot assessment of the participants’ experience in the co-creation process, the acceptability of the MOOCs and their effect on self-perceived DHL.

## 2. Materials and Methods

The methodological design carried out for the development of the MOOCs included a series of preliminary phases composed of a review of the literature, an exploratory survey with all cohorts of participants in the IC-Health project and the conduct of focus groups and group interviews, followed by the formation of communities of practice (CoPs) to co-design the MOOCs (Figure 1). A broader description of the methodology of IC-Health Project can be found in Perestelo-Pérez et al. 2020 [33]. In this paper, we report the results of the focus groups and the co-creation process whereby MOOCs aimed to PLW were developed.

The partner organizations were responsible for the necessary procedures to request the approval by the corresponding ethics committees of their organizations and they assured the research activities’ compatibility with national and European ethics requirements to protect the rights, safety and wellbeing of the participants involved.

### 2.1. Recruitment and Procedure

Participants were recruited through snowball sampling [34] from primary care centers, pregnancy and breastfeeding support groups, and social networks. The confidentiality of the patients’ personal data was guaranteed in accordance with the European Commission’s guidelines.

#### 2.1.1. Focus Groups

Focus groups with PLW were carried out in Spain and Italy between March and April 2017. A semi-structured guide was used to qualitatively explore the dimensions of DHL and to complement the information of the survey.

#### 2.1.2. Communities of Practice for the Development of MOOCs

For the co-creation of the MOOCs, a CoP [35,36] was created in each participating country (Belgium, Denmark, Italy, Spain and Sweden). Each CoP was composed of key stakeholders (PLW, their partners, researchers and interested healthcare professionals as midwives and lactation monitors) and coordinated by a CoP coordinator (research member of the project) (see Appendix A).

The co-creation process of the MOOCs started with a face-to-face group session that lasted approximately two hours in each country. In the first session, participants were informed in detail about the project and they signed the consent form. Then, a preliminary storyboard for the MOOC was defined regarding dimensions of DHL (finding, understanding, appraisal and applying health information) with examples in the context of pregnancy and lactation. At the end of this session, participants were encouraged to continue the process of co-creation online between October 2017 and April 2018, through a closed Moodle platform.

In all CoPs, at least one activity was presented weekly on the platform (e.g., review of a document, video, choice of images, suggestion of content, etc.). The feedback of the participants on the content and design of each MOOC was considered to conduct the pertinent modifications.

After the online phase, another round of face-to-face sessions was carried out, in which participants provided their final feedback about the ease of navigation, accessibility, duration, language, content and structure. In this session, a pilot quantitative assessment was performed and the sample that participated in the co-creation process were asked for their experience developing and using the MOOCs, with the aim of improving the co-creation process in future studies. All sessions followed a semi-structured guideline.

### 2.2. Measures

#### Quantitative Measures

Experience during the co-creation process, acceptability of the MOOCs and self-perceived DHL were assessed with the following questionnaire:−Experience during the co-creation process was assessed with 3 self-developed items, with a 4-point Likert scale from 0 (totally disagree) to 4 (totally agree) (see items in Section 3.2.1).−The acceptability of the MOOCs was assessed through a 3 open questions and 11-item questionnaire on a 3 or 4-point Likert scale developed specifically for the IC-Health project and based on previous related studies [37] that assessed ease of navigation, objectives and language clarity, appropriateness of learning activities and quizzes, and other characteristics of the MOOCs (see items in Section 3.2.2).−Self-perceived DHL was assessed before and after the MOOCs development. We used 8 items selected from 3 validated questionnaires: 5 items of the eHealth literacy scale (eHEALS) [38], 2 items of the eHealth impact questionnaire (eHIQ-Part 1) [39], and 2 item of the health literacy questionnaire (HLQ) [40] (see items in Section 3.2.3). Items had a response gradient from 0 (totally disagree) to 4 (totally agree) and assessed the three main skills required in DHL: finding (3 items), understanding (2 items) and appraisal (3 items) information on the Internet. The total score on these scales was divided by their corresponding number of items.

### 2.3. Analysis

#### 2.3.1. Qualitative Measures

The focus groups were audio recorded and analyzed by means of a descriptive deductive content analysis [41]: (1) in-depth analysis of the audio registration; (2) identification of relevant questions or issues discussed; (3) codification of each relevant topic; (4) grouping the information obtained on each topic; (5) critical analysis and interpretation of information collected on each explored topic; (6) incorporation of the observations recorded by the moderator and assistant; and (7) synthesis of the results.

#### 2.3.2. Quantitative Measures

Means and standard deviations were calculated for all items assessed (acceptability, DLH and experience during the co-creation workshops), and for the DHL scales (finding, understanding and appraising). For acceptability and experience, we also analyzed the response distribution for each item.

For self-perceived DHL, the following exploratory analyses were performed: differences between the three dimensions (finding, understanding and appraisal), both at baseline and after the MOOCs’ development/review, were compared with a paired *t*-test. Wilcoxon rank-sum test was used to analyze differences at baseline between sociodemographic groups: age (<29 vs. ≥30), education (university vs. high school or less), Southern (Spain, Italy) vs. Northern (Sweden, Denmark, Belgium) countries, as well as between each DHL subscale at both time points.

## 3. Results

### 3.1. Focus Groups

Seventeen PLW participated in focus groups (Table 1).

Experiences, needs, expectations and trust in the use of the Internet as a source of information on health and illness issues were the main themes identified (Table 2).

In general, the Internet was recognized as a source of secondary health information normally used to find quick answers to specific questions/problems when communication with the midwives is not possible. The participants did not find appropriate to use the Internet when healthcare professionals could be consulted beforehand, when the topic of interest was a serious health problem or when direct face-to-face interaction learning was required. The participants recommended using and contrasting different sources of information online; they would prefer to access official reliable information; and sometimes the access to information is conducted by another person who becomes a filter of the information.

Regarding usability, participants preferred resources, such as videos or images, as well as the use of non-technical language that facilitated understanding (see Appendix A).

### 3.2. Communities of Practice for the Development of MOOCs

A total of 237 people from 5 countries were invited to participate in the CoPs, of whom 113 accepted, the majority in Spain (72.6% of the total invited and 64.6% of participants) (Figure 2). Sixteen (14.2%) were male partners of PLW. Ninety-four (83.2%) were younger than 40 years old. Concerning education levels, 71% had university studies and 16.8% a high school degree. Most of the participants used the Internet every day (84.1%) or at least once per week (12.4%).

Overall, the online co-creation process worked well, since the planned MOOCs could be developed in all cases and the evaluation of the experience of the participants during the process showed favorable results (see the Quantitative Outcomes Section). However, during the last weeks of pregnancy till the first months after childbirth, the participation in face-to-face and online sessions was increasingly difficult, resulting in dropouts. In Spain, the country with more participants, many of them showed low participation, but the sample was large enough to enable the saturation of participants’ preferences and suggestions.

A total of six MOOCs were developed, two in Spain and one in the remaining countries. The Spanish participants chose to make two MOOCs (one with basic information and the other with more advanced information on skills) to better meet their information needs throughout the pregnancy and postpartum process. Initially, MOOCs were planned for a maximum duration of 15 min. However, by including the extra materials and resources requested by the participants, the average duration to complete and visualize entirely the materials was around 60–90 min. Nevertheless, this Appendix A did not require mandatory consultation to achieve an effective knowledge of each skill; it was complementary information that supported users to expand the information if considered necessary.

The structure and format of the materials of each MOOC were adapted to the interests of the PLW in each country, but all of them included an initial description of the course, followed by an introductory unit on DHL and four more units, one for each skill explained (find, understand, appraisal and apply). A definition of each skill was offered, as well as strategies to put them into practice in the areas of pregnancy, childbirth and/or breastfeeding. These strategies were accompanied by practical examples presented in various formats (videos, infographics, PDFs, etc.). For example, in the Spanish MOOCs for find, tips were offered to plan Internet searches and perform advanced searches on concepts related to pregnancy, childbirth and breastfeeding. For understand, recommendations were offered to improve reading comprehension of obstetrician medical terminology about pregnancy and breastfeeding. For appraisal, tips were offered to quickly recognize reliable web pages and strategies to evaluate in depth the compliance with the Health on the Net Foundation Code of Conduct (HONcode) [42] for medical and health web sites or the accredited medical website (WMA) [43], among others. For apply, some recommendations were offered regarding different ways to apply the online information (e.g., share it in consultations with the midwife or lactation monitor and facilitate shared decision making or to improve self-care in daily life). Units also include self-assessment questions on the contents debated in each of them.

The MOOCs developed were uploaded on a Moodle platform and an updated version of the Spanish MOOCs can be found on the University of La Laguna’s website [44] (see Appendix A).

#### 3.2.1. Experience during the Co-Creation Process

Data were available for 76 participants (Figure 2). The percentage of patients who totally agreed or agreed with item 1 (*“**Being part of the co-creation process made the MOOC content more relevant to my needs”)* was 85.5% (mean = 2.99, sd = 0.66); for item 2 (*“**The co-creation process made me feel part of the project”*), it was 89.5% (mean = 3.06, sd = 0.62); and 82.9% (mean = 3.04, sd = 0.77) for item 3 (*“**Taking part in the different workshops has improved my knowledge about digital health literacy. This has increased my ability to take charge of my health”*).

#### 3.2.2. Acceptability of the MOOCs

Acceptability data were available for 68 participants (Table 3). Combining the “totally agree” and “agree” categories, most of them thought that the language on the MOOC was easy to understand (94.1%), the objectives of the course were clear (98.5%), and the contents were consistent with them (92.2%). More than 80% perceived that the learning activities were useful, and quizzes appropriately tested the material presented. The MOOC was easy to navigate for 76.5%, and 72.1% thought that the quality of the overall design and materials was high or very high. Ninety percent felt that their expectations were met and 76.5% would recommend it to other people. Examples used in MOOCs to exemplify each skill received the lowest acceptance rates, with participants similarly distributed across response categories (*high quality*, *low quality* and *not sure*).

#### 3.2.3. Self-Perceived DHL

Baseline data were available for all participants included in the co-creation process (*n* = 113). Mean scores at the item level (theoretical range 0–4) were between 2 and 3 for all of them (Table 4). Mean scores in the three dimensions were 2.59 (sd = 0.74) (finding), 2.56 (0.78) (understanding) and 2.44 (0.73) (appraisal). This latter dimension significantly differed from the other two (*p* = 0.033 and *p* = 0.042, respectively). Participants with university degrees scored significantly higher than those with high school or less in understanding (z = −2.34, *p* = 0.019); the *p*-value for appraisal was 0.057 in the same direction. Southern countries showed significantly worse scores in understanding (z = −2.35, *p* = 0.018) and appraisal (z = −2.32, *p* = 0.020); finding obtained a *p*-value of 0.054.

Data after reviewing the definitive versions of the MOOCs were available for 98 participants. Means were above 2.80 in all items, and above 3 in 5 of them (Table 4). Total scores were 3.01 (0.58) for finding, 3.04 (0.62) for understanding and 2.96 (0.64) for appraisal (*p* = 0.046 versus understanding; the remaining differences were not significant). Southern countries showed worse scores in the three dimensions, but the difference only was significant for appraisal (z = −3.27, *p* = 0.001). Understanding obtained a *p*-value of 0.059.

Comparison between pre–post results yielded significant results for all items except for the second item of finding *(“I get nervous using the Internet to find information about my health”)*, and for the three dimensions (*p*-values < 0.003) (Table 4) with better scores after the development/review of the MOOCs.

## 4. Discussion

The primary aim of this project was to develop MOOCs oriented towards the improvement of the DHL of European PLW, using a co-creation process that makes the final product respond to the needs and interests of the targeted participants. Previous investigations have shown that stakeholders’ involvement in the design of health interventions improves their perceived quality of the developed materials or procedures [45], and that co-creation can help to improve some health-related outcomes and often lead to a better assessment of their relevance [46,47]. Although quantitative results are preliminary, most participants in this study positively value both their participation in the process, and the different quality dimensions of the MOOCs developed, being similar to those observed in other studies [48]. Nonetheless, the results showed that several aspects could still be improved, such as the complexity of the examples used to exemplify each DHL skill, or navigation issues. It is reasonable to expect targeted users to be more accepting of short educational resources; however, the results show that PLW requested and positively valued the possibility of expanding the information presented in the MOOCs, thus increasing their final duration. The results presented on self-perceived DHL are also preliminary and exploratory, since the evaluation of the effectiveness of the MOOCs was not the main objective of the project. However, according to a recent systematic review on MOOC evaluation methods, the adoption of a mixed methods analysis that considers both quantitative and qualitative data may be more useful to evaluate the overall quality of MOOCs, since it allows a better understanding of the generated metrics and produces greater knowledge for the future improvement of MOOCs [49]. The assessed correlates of the DHL dimensions showed expected results, with lower scores in less educated participants, and in Spanish and Italian ones compared to those from Northern countries, which have a greater socioeconomic development. Social and demographic factors, such educational level, culture and language, influence HL skills [50,51,52] observing that a low socio-educational level can be a risk factor for a low level in HL. Considering this relationship, it is not surprising to observe that the lowest health-related skills are found among people with the most disadvantaged socioeconomic status [53]. Nevertheless, HL levels then become a modifiable risk factor for health disparities on which it is possible to act to reach greater health equity [53]. After the use of MOOCs, scores were significantly higher in the three dimensions, and differences between them were lower than at baseline. As previously mentioned, these results are exploratory and subject to several limitations, and a rigorous assessment of the effectiveness of the MOOCs for the improvement of self-perceived and objective DHL is needed.

Previous interventions to improve DHL using a design similar to this study have shown that digital health technology facilitates access to health care and HL, providing opportunities to increase the scope and participation of patients in these types of interventions [54,55].

Regarding the use of the Internet for health topics, previous studies show that those PLW who ask questions and compare information with their healthcare professionals and jointly choose the best plan of care and make shared decisions about their health tend to be more satisfied with the care they receive, as they gain autonomy in managing their health [1,56,57,58]. Promoting the empowerment of PLW also can be a catalyst for a better health self-management, leading to better outcomes for the mother and the child. Different studies have shown that lower HL is negatively related to the ability to evaluate and trust online health information [59]. Moreover, poor HL skills are also associated with lower odds of using and trusting healthcare professionals and medical websites and with higher odds of using low-quality information obtained from television, social media and blogs without verified information [60]. An important aspect when using social media and blogs as information resources is to evaluate their suitability for the individual characteristics of the user. A recent literature review highlights the importance of using social media in the healthcare industry and raises the need for health organizations develop a strategic plan for digital communication, considering the impact of social networks have as a powerful tool for empowering people by providing informational and emotional support [61].

Therefore, it is essential to promote the development of competencies that allow PLW to assess the quality and accuracy of online health information in order to make informed decisions about their health, particularly when access to health care is limited [62,63]. The use of resources such as MOOCs with interactive tools, can be a possibility of shared empowerment between healthcare professionals and share similar experiences with other PLWs in the same situation [64]. This combination of passive learning with interactive tools of social networks can help PLW to improve their self-information skills, therefore healthcare professionals must be prepared to support the retrieval, interpretation and request of online information [1,47,65,66]. A recent systematic review aimed at understanding midwives’ perspectives on women self-monitoring of their pregnancy using eHealth concluded that they generally hold ambivalent views [67]. While noting its potential to help women to make informed decisions, they also point out the risk of accessing inaccurate information [67]. This highlights the necessity of providing PLW with specific skills to find and interpret valid and reliable health information online. Furthermore, aspects, such as ease of access from anywhere, at any time and from any device, contemplated in this work, favor that MOOCs can be adapted to the own pace and time needs of the obstetric population. In this sense, the IC-Health MOOCs for PLW can be a useful resource on maternal and child health websites, health centers and health-care programs for childbirth and breastfeeding.

### Strengths and Limitations

The main strength of this project is the implication of the target population in the development of the MOOCs to improve the relevance of the contents included and the way they are presented. Good acceptability of educational resources is a necessary requisite for their effectiveness in the improvement of DHL, and currently there is an increasing interest in involving patients and the general population in the development of interventions aimed at improving their health, in order to maximize the effectiveness and efficiency of such interventions. The developed MOOCs contribute to this aim in conditions of such relevance to public health as pregnancy and lactation. On the other side, the international nature of the project contributes to homogenize methodologies across European research groups but respecting the interests and needs of the target audience in each country.

Nonetheless, there are several limitations to the study. There were few participants in Belgium and Italy, and therefore, the materials developed in these two countries possibly do not represent all the preferences and interest of PLW, although in the case of Italy at least one focus group could be carried out before the formation of the CoP. On the contrary, two-thirds of the total sample who participated in the co-creation process were Spanish, and the quantitative results therefore mainly represent this population. Many of them participated scarcely, but the number of participants was great enough to reach saturation of preferences and suggestions. Maternity is a life-changing experience and socio-cultural differences influence the way women understand both pregnancy and lactation [68,69]. This study was carried out in a European population, so these MOOCs could require a cultural adaptation of the perspective of pregnancy and lactation in other non-European countries.

Another limitation is that due to confidentiality issues we could not analyze separately the participants who participated in the co-creation process and those who only reviewed the final version of the MOOCs. It is possible that participants who did not participate in the co-creation process showed less acceptability, and it could be the cause of the lower ratings attributed to the examples used in the MOOCs to exemplify each DHL skill. Therefore, an independent assessment of acceptability is still needed to confirm the results obtained. The same limitation applies to the exploratory assessment of DHL, so that the MOOCs’ effectiveness must be evaluated by means of a randomized clinical trial.

## 5. Conclusions

The work carried out on this project with European PLW provides an example of the MOOC co-creation process to promote DHL in this population. The preliminary results are encouraging to drive the generation of more digital resources of this type for topics related to self-care in pregnancy and postpartum that could be offered in childbirth preparation classes and breastfeeding support classes in European health services. However, studies focusing on evaluating the impact of these interventions are necessary to capture the real change in women’s DHL level, as well as other variables, such as knowledge of pregnancy and lactation, decisional conflict about medical procedures, or satisfaction.

## Figures and Tables

**Figure 1 ijerph-19-00913-f001:**
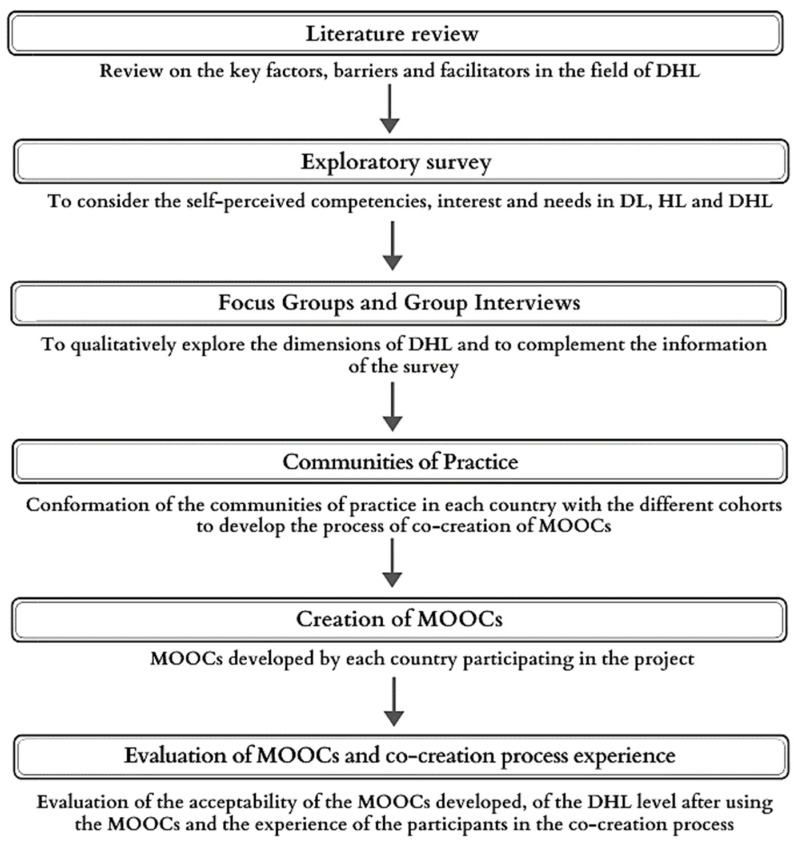
Procedure of IC-Health project.

**Figure 2 ijerph-19-00913-f002:**
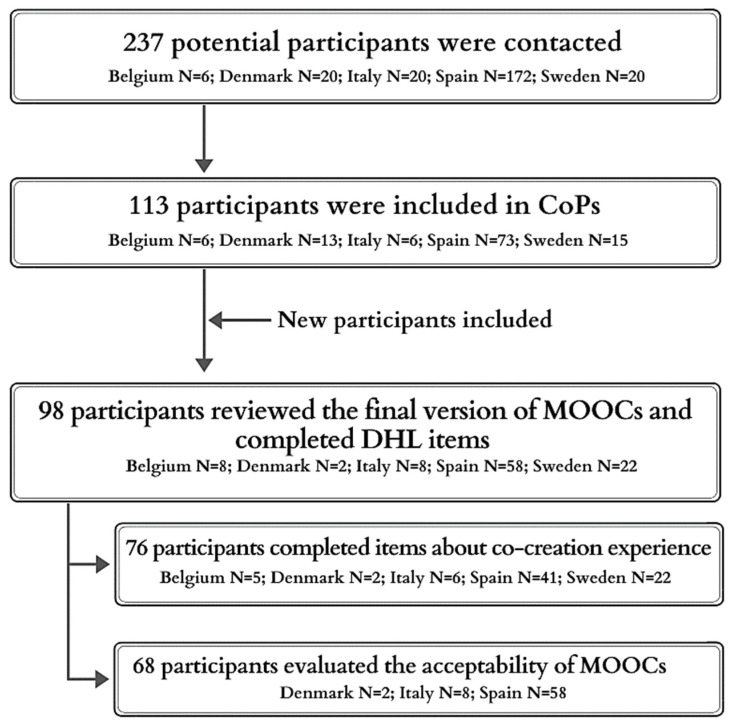
Flow diagram of participation in the co-creation process.

**Table 1 ijerph-19-00913-t001:** Characteristics of the participants in Focus Groups.

Participants	Total Pregnant and Lactating Women	Pregnant Women	Lactating Women
Country (*n*)	17	8	9
Spain	11	7	4
Italy	6	1	5
Age range (years)	26–41	28–39	26–41
Spain	26–40	28–40	26–40
Italy	38–41	39	38–41
Education (*n*)			
University Degree	12	3	9
High School	3	-	3
Vocational superior training	2	2	-
Civil status (*n*)			
Married/Living with partner	14	5	9
Single	3	-	3
Employment Status (*n*)			
Employed	13	6	7
Unemployed	3	1	2
Retired	1	1	-
Occupation (*n*)			
Office work	6	1	5
Intellectual scientific work	4	2	2
Technician	4	3	1
Entrepreneur/executive	1	1	-
Dealer/trader	2	1	1

**Table 2 ijerph-19-00913-t002:** Themes and subthemes identified in the thematic analysis.

Themes	Subthemes	Example Quote
Experience/general opinion using the Internet for health and illness issues	− Personal experiences − Level of satisfaction − Use of this information	*“Before pregnancy, I had never thought about the Internet and health as connected. When I first got pregnant, I started using Internet for health-related issues because I was curious to understand what was happening to my body, since I do not know anything about medicine”* *“When doctors are not available or when the human touch is missing, Internet can compensate this lack of empathy by health personnel”* *“I have searched on the Internet (…) in other cases I say to myself that just in case I should go to the pediatrician”*
Needs and expectations of the use of the Internet as a source of information on health and illness issues	− Informational needs − Preferences relating display format	*“Issues like ante-natal courses and preparation to delivery cannot be fully understood if read… You need sharing views with other peers, to listen to midwives’ indications, to make questions… maybe a webinar can help, but certainly websites are not enough”* *“If doctors talk too technical, I don’t understand. I think if the information is presented in videos or images, it is easier for me to understand it”*
Trust on the Internet as a source of information on health and illness issues	− Situations of NOT using − Why you trust information − Issues enhance or diminish level of trust	*“I refused to look on the Internet about vaccines. I know the web is full of fake news about it and I simply didn’t want to assist or take part in that debate”* *“I tend to verify the source of information… I look for sites linked to health institutions, health professionals’ orders, research centers… I link to science-based sites”* *“I have always been a bit skeptical about looking for health information on the web, also listening to some friends that joined mothers’ forums where information is not filtered… They told me it just made them more confused and anxious”*
Perception of the use of the Internet as a source of information on health and illness issues by other people	− What use − Recommend other people	*“People who suffer from serious diseases and want to escape from reality, commonly look on the Internet to find a way out of their situation”* *“My pediatrician looks completely disconnected... I think she’s a very good doctor but, since she’s completely out of the world of Internet, instead of considering it a supportive tool she takes it as an enemy. If she was keener to suggest some online readings to mothers who bother her with minor babies’ health issues, maybe her workload could decrease…. but she does not even have an email”*

**Table 3 ijerph-19-00913-t003:** Acceptability of the MOOCs (*n* = 68).

Questions	Totally Agree *n* (%)	Agree *n* (%)	Not Sure *n* (%)	Disagree *n* (%)	Totally Disagree *n* (%)	Mean ^1^ (sd)
1. The MOOC is easy to use/navigate and the information was clearly organized	15 (22.1)	37 (54.4)	6 (8.8)	6 (8.8)	4 (5.9)	2.78 (1.08)
2. The language on the MOOC was easy to understand	18 (26.5)	46 (67.6)	3 (4.4)	0 (0.0)	1 (1.5)	3.18 (0.64)
3. The objectives of the course were made clear	18 (26.5)	49 (72.1)	0 (0.0)	0 (0.0)	1 (1.5)	3.22 (0.59)
4. The course content was consistent with the course objectives	20 (29.4)	42 (61.8)	2 (2.9)	3 (4.4)	1 (1.5)	3.13 (0.79)
5. The learning activities were useful to gain a clear understanding of the course content	16 (23.5)	42 (61.8)	8 (11.8)	1 (1.5)	1 (1.5)	3.04 (0.74)
6. The quizzes did appropriately test the material presented in the course	12 (17.6)	43 (63.2)	11 (16.2)	1 (1.5)	1 (1.5)	2.94 (0.73)
7. This course has met my expectations	12 (17.6)	49 (72.1)	3 (4.4)	3 (4.4)	1 (1.5)	3.00 (0.73)
8. I would recommend this course to other people	25 (36.8)	27 (39.7)	13 (19.1)	1 (1.5)	2 (2.9)	3.06 (0.94)
	High or very high *n* (%)		Not sure *n* (%)	Low *n* (%)	Very low *n* (%)	Mean ^1^ (sd)
9. Quality of the overall design and aesthetics of the contents and materials	49 (72.1)		16 (23.5)	3 (4.4)	0 (0.0)	2.67 (0.56)
10. Quality/usefulness of the examples provided in the course	23 (33.8)		24 (35.3)	19 (27.9)	2 (2.9)	2.00 (0.86)
	Yes *n* (%)		Too short *n* (%)		Too long *n* (%)	
11. Was the amount of time appropriate for the course content?	53 (77.9)		7 (10.3)		8 (11.8)	
12. Open question: Please provide a short summary of the strengths and weaknesses of the course						
13. Open question: Please provide brief suggestions on how to improve the course						
14. Open question: What are the main points that you have learned through this course?						

^1^ Higher scores indicate more positive rating (range 0–4 for items 1–8, and 0–3 for items 9 and 10).

**Table 4 ijerph-19-00913-t004:** Digital Health scores.

Digital Health Literacy Items ^a^	Baseline Sample (*n =* 113)	Post Sample (*n =* 98)	z (*p*) ^b^
F1. I know how to find useful health resources on the Internet	2.62 (0.94)	3.14 (0.62)	−4.56 (<0.001)
F2. I get nervous using the Internet to find information about my health (reversed)	2.62 (1.04)	2.82 (0.95)	−0.99 (0.320)
F3. I know where to find useful health resources on the Internet	2.54 (0.91)	3.07 (0.65)	−4.69 (<0.001)
Finding total	2.59 (0.74)	3.01 (0.58)	−4.68 (<0.001)
U1. I know how to use the Internet to help me to understand what I am not sure about my health	2.59 (0.92)	3.03 (0.63)	−3.74 (<0.001)
U2. I can understand the health information I get from the Internet well enough to know what to do	2.53 (0.91)	3.05 (0.72)	−4.70 (<0.001)
Understanding total	2.56 (0.78)	3.04 (0.62)	−3.93 (<0.001)
A1. I have the skills I need to evaluate the health resources I find on the Internet	2.62 (0.84)	2.96 (0.72)	−2.98 (0.003)
A2. I can differentiate high-quality health resources from low-quality health resources on the Internet	2.66 (0.81)	3.11 (0.66)	−4.47 (<0.001)
A3. I feel confident in using information from the Internet to make health decisions	2.04 (0.96)	2.81 (0.84)	−5.67 (<0.001)
Appraising total	2.44 (0.73)	2.96 (0.64)	−5.68 (<0.001)

^a^ Higher score is better; ^b^ z (*p*-value) from Wilcoxon rank-sum tests.

## Data Availability

The data presented in this study are available in Appendix A.

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
