# Peer review of "Co-Creation of Massive Open Online Courses to Improve Digital Health Literacy in Pregnant and Lactating Women"

_ijerph, 2022, doi:10.3390/ijerph19020913_

Round 1
Reviewer 1 Report
Thank you for inviting me to review this article titled “Co-creation of Massive Open Online Courses to improve Digital Health Literacy in Pregnant and Lactating Women”. The study presents some of the results got within a European Project IC-Health: Improving digital health literacy in Europe, with specific regard to the focus group/interviews and communities of practices that supported the creation of MOOCs target to pregnant and lactating women in order to improve their DHL.
For method, authors should provide with more information about items used as quantitative measures. They should show which are the items used for “experience” and “acceptability” as well as whether and how the self-developed items have been validated by authors. Then, as for acceptability, in the method section, you indicated you used 14 items, but in findings I see only 11 items (table 3). Justify.
The study presents interesting findings about the value of adopting a co-creation process to improve the DHL of European PLW. Despite most participants are Spanish, I would suggest to discuss more the role of the socio-cultural differences among participants in focus groups and communities of practices (e.g. the different cultural approach to pregnancy showed in different countries) and their familiarity with technology which can be an enabler or obstacle to take advantage of MOOCs (digital divide is a relevant theme for the EU).
Another point worth of investigation could be the combination of the passive learning with interactive tools enabling pregnant and lactating women to interact with health-care providers and/or other women experiencing the same conditions. In this regard, “Pianese, T., & Belfiore, P. (2021). Exploring the Social Networks’ Use in the Health-Care Industry: A Multi-Level Analysis. International Journal of Environmental Research and Public Health, 18(14), 7295.” provide with a literature review about the opportunities from using social media in the healthcare industry. Interactive forum and chats, along with social networks have been demonstrated to be a powerful tool to empower people by providing them with informational and emotional support.
Hope these comments would be valuable for the authors.
*. This reviewer shared the same affiliation with one of the co-authors but she/he had no association with the study described in the reviewed manuscript or received funding from it, and therefore declared no conflict of interest.
Author Response
Point 1: Thank you for inviting me to review this article titled “Co-creation of Massive Open Online Courses to improve Digital Health Literacy in Pregnant and Lactating Women”. The study presents some of the results got within a European Project IC-Health: Improving digital health literacy in Europe, with specific regard to the focus group/interviews and communities of practices that supported the creation of MOOCs target to pregnant and lactating women in order to improve their DHL.
For method, authors should provide with more information about items used as quantitative measures. They should show which are the items used for “experience” and “acceptability” as well as whether and how the self-developed items have been validated by authors. Then, as for acceptability, in the method section, you indicated you used 14 items, but in findings I see only 11 items (table 3). Justify.
Response 1: Thank you very much for your comments. We have indicated that the Results show the items for each qualitative measure. In relation to validation, the experience during the co-creation process and the acceptability of the MOOCs were assessed through self-developed items for this study (i.e., these measures have not been validated). To date, they are not psychometric scales, that is, they are not designed to assess a latent construct (there is not a total score) and inter-item correlations are not relevant to our purpose, but the means/percentages obtained in each of the items. Questions to explore the experience during the co-creation process were also explored through group discussion on the effectiveness and quality of the co-creation process in face-to-face sessions with the participants (explained in the manuscript). Acceptability items were based on previous related studies (references are now cited in the manuscript). Regarding the number of items for acceptability, this is a misprint in the manuscript that we have already corrected: there were 3 open questions and 11 items (14 questions in total).
Point 2: The study presents interesting findings about the value of adopting a co-creation process to improve the DHL of European PLW. Despite most participants are Spanish, I would suggest to discuss more the role of the socio-cultural differences among participants in focus groups and communities of practices (e.g. the different cultural approach to pregnancy showed in different countries) and their familiarity with technology which can be an enabler or obstacle to take advantage of MOOCs (digital divide is a relevant theme for the EU).
Response 2: Thank you for your appreciation. The participants in this study showed sociodemographic similarities (71% had university studies and 84.1% used the Internet every day). However, although in this study no socio-cultural differences were observed in the discourse of the participants (in Focus Groups) or in their contributions during the process of co-creation of MOOCs (in the Communities of Practices), we think it is important to point out the possible influence that the sociocultural perspective on pregnancy and lactation could have on the development of MOOCs, in non-European contexts. Therefore, we have discussed this aspect in the Limitations section. Regarding digital divide, we have also added in Introduction section information regarding the relevance in the identification of some gaps related to DHL PLW levels by the European Commission reflected in the "European citizens' digital health literacy" report of the Flash Eurobarometer 404.
Point 3: Another point worth of investigation could be the combination of the passive learning with interactive tools enabling pregnant and lactating women to interact with health-care providers and/or other women experiencing the same conditions. In this regard, “Pianese, T., & Belfiore, P. (2021). Exploring the Social Networks’ Use in the Health-Care Industry: A Multi-Level Analysis. International Journal of Environmental Research and Public Health, 18(14), 7295.” provide with a literature review about the opportunities from using social media in the healthcare industry. Interactive forum and chats, along with social networks have been demonstrated to be a powerful tool to empower people by providing them with informational and emotional support.
Hope these comments would be valuable for the authors.
Response 3: Thank you very much for your comments and for providing us with such this interesting reference. We have added the reference and highlighted the importance of using social media in the healthcare industry in Discuss section.

Reviewer 2 Report
Thanks for giving me the chance to review this very interesting article that I am sure will make an important contribution to the broader field of health literacy.
The work is very well planned and very well articulated, both from a theoretical and methodological perspective.
However, there are two important flaws that should be addressed before publication. I am sure that the author(s) will be able to solve them very easily.
- Theoretical framework
- a) In the Introduction author (s) provide an overview of digital health literacy but surprisingly only dedicate one quick sentence to MOOCs.
Since the article is about the creation of a MOOC, it is seminal to dedicate at least one paragraph to 1) Explain what Moocs are 2) Discuss relevant literature in the field 3) use the previous to summarize what has been done until now in terms of MOOC evaluation.
This is seminal to locate your research in the broader field of studies of MOOC, to increase the paper'sinterest and citability and mlost importantly justify all your analysis.
This reference might help contextualize MOOCs:
Cervi, Laura, José M. Pérez Tornero, and Santiago Tejedor. 2020. "The Challenge of Teaching Mobile Journalism through MOOCs: A Case Study" Sustainability 12, no. 13: 5307. https://doi.org/10.3390/su12135307
CONCLUSIONS
Once the literature review has been properly discussed the Conclusion should allow your results to “dialogue” with previous/similar studies allowing to state 1) what is your contribution 2) what are the limitations of this study 3) offering suggestions on how these results can guide future education plans.
Author Response
Point 1: Thanks for giving me the chance to review this very interesting article that I am sure will make an important contribution to the broader field of health literacy.
The work is very well planned and very well articulated, both from a theoretical and methodological perspective.
However, there are two important flaws that should be addressed before publication. I am sure that the author(s) will be able to solve them very easily.
- Theoretical framework
- In the Introduction author (s) provide an overview of digital health literacy but surprisingly only dedicate one quick sentence to MOOCs.
Since the article is about the creation of a MOOC, it is seminal to dedicate at least one paragraph to 1) Explain what Moocs are 2) Discuss relevant literature in the field 3) use the previous to summarize what has been done until now in terms of MOOC evaluation.
This is seminal to locate your research in the broader field of studies of MOOC, to increase the paper'sinterest and citability and mlost importantly justify all your analysis.
This reference might help contextualize MOOCs:
Cervi, Laura, José M. Pérez Tornero, and Santiago Tejedor. 2020. "The Challenge of Teaching Mobile Journalism through MOOCs: A Case Study" Sustainability 12, no. 13: 5307. https://doi.org/10.3390/su12135307
Response 1: Thank you very much for your comments. The reference you provide has helped us to relate the results of our study with similar experiences and we have expanded the description of MOOCs in the Introduction section, as you suggested. We have also discussed the importance of the use of mixed methodologies to evaluate the overall quality of MOOCs and we showed how our results are related to previous studies in the field (Discussion section).
Point 2: CONCLUSIONS. Once the literature review has been properly discussed the Conclusion should allow your results to “dialogue” with previous/similar studies allowing to state 1) what is your contribution 2) what are the limitations of this study 3) offering suggestions on how these results can guide future education plans.
Response 2: In section “4.1. Strengths and Limitations” of the Discussion we comment that the international nature of the project contributes to homogenize methodologies across European research groups respecting the interests and needs of the target audience in each country; and we propose that one way to evaluate the effectiveness of MOOCs (which could not be done in this study and involves a limitation) is through a randomized clinical trial. Additionally, we have added some suggestions in Conclusion section to include MOOCs among the resources offered to PLWs in childbirth or breastfeeding support classes in different European Health Services.

Round 2
Reviewer 2 Report
publishable